# An integrated clinical and genetic model for predicting risk of severe COVID-19: A population-based case–control study

**Gillian S. Dite**[ID]*☯, **Nicholas M. Murphy**☯, **Richard Allman**

Genetic Technologies Ltd., Fitzroy, Victoria, Australia

☯ These authors contributed equally to this work.
* gillian.dite@gtglabs.com

**Data Availability Statement:** Access to the data used in this study can be obtained by applying directly to the UK Biobank at https://www.ukbiobank.ac.uk/register-apply/. The authors did not receive special access privileges to the data

## Abstract

Up to 30% of people who test positive to SARS-CoV-2 will develop severe COVID-19 and require hospitalisation. Age, gender, and comorbidities are known to be risk factors for severe COVID-19 but are generally considered independently without accurate knowledge of the magnitude of their effect on risk, potentially resulting in incorrect risk estimation. There is an urgent need for accurate prediction of the risk of severe COVID-19 for use in workplaces and healthcare settings, and for individual risk management. Clinical risk factors and a panel of 64 single-nucleotide polymorphisms were identified from published data. We used logistic regression to develop a model for severe COVID-19 in 1,582 UK Biobank participants aged 50 years and over who tested positive for the SARS-CoV-2 virus: 1,018 with severe disease and 564 without severe disease. Model discrimination was assessed using the area under the receiver operating characteristic curve (AUC). A model incorporating the SNP score and clinical risk factors (AUC = 0.786; 95% confidence interval = 0.763 to 0.808) had 111% better discrimination of disease severity than a model with just age and gender (AUC = 0.635; 95% confidence interval = 0.607 to 0.662). The effects of age and gender are attenuated by the other risk factors, suggesting that it is those risk factors–not age and gender–that confer risk of severe disease. In the whole UK Biobank, most are at low or only slightly elevated risk, but one-third are at two-fold or more increased risk. We have developed a model that enables accurate prediction of severe COVID-19. Continuing to rely on age and gender alone (or only clinical factors) to determine risk of severe COVID-19 will unnecessarily classify healthy older people as being at high risk and will fail to accurately quantify the increased risk for younger people with comorbidities.

## Introduction

The current COVID-19 pandemic is a dominating and urgent threat to public health and the global economy. While COVID-19 can be a mild disease in many individuals, with cough and fever the most commonly reported symptoms, up to 30% of those affected may require hospitalisation, and some will require intensive intervention for acute respiratory distress syndrome [1, 2].

that others would not have; Interested researchers will be able to access the data in the same manner by applying directly to the UK Biobank.

**Funding:** The authors received no specific funding for this work. All authors are employed by a commercial company, Genetic Technologies Limited, which provided support in the form of salaries for all authors, but did not have any additional role in the study design, data collection and analysis, decision to publish, or preparation of the manuscript. The specific roles of all authors are articulated in the Author Contributions section.

**Competing interests:** I have read the journal's policy and the authors of this manuscript have the following competing interests: All authors are employed by Genetic Technologies Limited and have a patent pending (AU_2020901739 – Methods of assessing risk developing a severe response to Coronavirus infection) for the work in this manuscript. A product to predict risk of severe COVID-19 is in development. This does not alter our adherence to PLOS ONE policies on sharing data and materials.

Globally, public health responses have been aimed at limiting new cases by preventing community transmission through mask wearing, social distancing, curtailing non-essential services and broad travel restrictions. The economic and social impacts of these interventions have been devastating, with foundational damage to local economies [3] and unprecedented increases in mental health diagnoses being reported [4].

As the protracted strain of the pandemic increases pressure to re-open economies, there is an urgent need for tests to predict an individual's risk of severe COVID-19. In the community, a risk prediction test could enable workplaces to confidently manage employees who are at increased risk of severe disease and should work from home or avoid client-facing roles. In the healthcare setting, a risk prediction test could inform patient triage when hospital resources are limited and be useful in prioritising pathology tests and vaccination (when one becomes available). On a personal level, knowledge of individual risk can empower individuals to make informed choices about day-to-day activities.

Age, gender, and comorbidities are frequently cited as risk factors for severe COVID-19 [5], but these have generally been considered independently without accurate knowledge of the magnitude of their effect on risk, potentially resulting in incorrect risk estimation. Early epidemiological analyses of the factors associated with COVID-19 severity and death have now appeared, including an analysis of a cohort of 17 million people by Williamson et al. [6] and a prospective cohort study of 5,279 people in New York [7], both based on the analysis of electronic health records.

The analysis of human genetic variation that may affect response to viral infection has been slower, largely due to the lack of available data. Nevertheless, the COVID-19 Host Genetics Initiative has undertaken meta-analyses of the genetic determinants of COVID-19 severity and has made the summary statistics publicly available [8, 9]. In addition, Ellinghaus et al. [10] identified two loci (3p21.31 and 9q34.2) that are strongly associated with severe disease.

We used the UK Biobank to develop a comprehensive model to predict risk of severe COVID-19 by integrating demographic information, comorbidity risk factors, and a panel of genetic markers.

## Methods

### UK Biobank data

The UK Biobank is a population-based prospective cohort of over 500,000 participants from England, Wales, and Scotland who were aged 40 to 69 years when recruited from 2006 to 2010 [11]. The UK Biobank has extensive genotyping [12] and phenotypic data obtained from baseline assessment and from linkage to hospital and primary care databases and to cancer and death registries [11]. The UK Biobank has Research Tissue Bank approval (REC #11/NW/0382) that covers analysis of data by approved researchers. All participants provided written informed consent to the UK Biobank before data collection began. This research has been conducted using the UK Biobank resource under Application Number 47401.

In response to the COVID-19 pandemic, the UK Biobank made available up-to-date SARS-CoV-2 testing, hospital, primary care, and death data for use in COVID-19 research by approved researchers [13]. We extracted testing and hospital records from the UK Biobank COVID-19 data portal on 15 September 2020. We extracted single-nucleotide polymorphism (SNP) and baseline assessment data from files previously downloaded as part of our approved project. At the time of data extraction, primary care administrative data (general practitioner records relating to diagnoses, symptoms, referrals, laboratory test results and prescriptions for medication) was only available for just over half of the identified participants and was therefore not used in these analyses.

## Eligibility

Eligible participants were those who had tested positive for SARS-CoV-2 and for whom SNP genotyping data and linked hospital records were available. Of the 18,221 participants with SARS-CoV-2 test results, 1,713 had tested positive and 1,582 of those had both SNP and hospital data available.

## COVID-19 severity

We used source of test result as a proxy for severity of disease: outpatient representing non-severe disease and inpatient representing severe disease. For participants with multiple test results, we considered the disease to be severe if at least one result came from an inpatient setting.

## Selection of SNPs for risk of severe COVID-19

We identified 62 SNPs from the results of the ANA2 meta-analysis (release 2) of SARS-CoV2 positive non-hospitalised versus hospitalised cases of COVID-19 conducted by the COVID-19 Host Genetics Initiative consortium [8, 9]. Because of the limited amount of data available at the time of release, we used $P<0.0001$ as the threshold for loci selection. We then removed variants that were associated with hospitalisation in only one of the five studies in the meta-analysis. We pruned for linkage disequilibrium using an $r^2$ threshold of 0.5 against the 1000 Genomes European populations (CEU, TSI, FIN, GBR and IBS) representing the ethnicities of the submitted populations [14]. Variants that had a minor allele frequency of $\geq 0.01$ and beta coefficients from −1 to 1 were then retained [15]. All of the variants had a Cochran's Q heterogeneity test $P>0.001$. Where possible, SNP variants were chosen over insertion–deletion variants to facilitate laboratory validation testing. We also included the two lead SNPs from the loci found by Ellinghaus et al. [10] that reached genome-wide significance. Therefore, we used a panel of 64 SNPs for severe COVID-19 in this study (S1 Table).

## SNP score

While we would normally construct a SNP relative risk score using published data to calculate population-averaged risk values for each SNP and then multiplying the risks for each SNP [16], the size of the odds ratios for the 64 SNPs meant that this approach could result in relative risks of several orders of magnitude. Therefore, for this study, we calculated the percentage of risk alleles present in the genotyped SNPs for each participant. We used the percentage rather than a count because some of the eligible participants had missing data for some SNPs (9% had all SNPs genotyped, 21% were missing 1 SNP, 26% were missing 2 SNPs, 18% were missing 3 SNPs, 10% were missing 4 SNPs, 7% were missing 5 SNPs, 8% were missing 6–10 SNPs and 1% were missing 11–15 SNPs).

## Imputation of ABO genotype

Blood type was imputed for genotyped UK Biobank participants using three SNPs (rs505922, rs8176719 and rs8176746) in the ABO gene on chromosome 9q34.2. A rs8176719 deletion (or for those with no result for rs8176719, a T allele at rs505922) indicated haplotype O. At rs8176746, haplotype A was indicated by the presence of the G allele and haplotype B was indicated by the presence of the T allele [17, 18].

## Clinical risk factors

Risk factors for severe COVID-19 were identified from large epidemiological studies of electronic health records [6, 7] and advice posted on the Centers for Disease Control and Prevention website [19]. Rare monogenic diseases (thalassemia, cystic fibrosis and sickle cell disease) were not included in these analyses.

Age was classified as 50–59 years, 60–69 years and 70+ years. This was based on the participants' approximate age at the peak of the first wave of infections (April 2020) and was calculated using the participants' month and year of birth. Self-reported ethnicity was classified as white and other (including unknown). The Townsend deprivation score at baseline was classified into quintiles defined by the distribution in the UK Biobank as a whole. Body mass index and smoking status were also obtained from the baseline assessment data. Body mass index was inverse transformed and then rescaled by multiplying by 10. Smoking status was defined as current versus past, never or unknown. The other clinical risk factors were extracted from hospital records by selecting records with ICD9 or ICD10 codes for the disease of interest (S2 Table).

## Statistical methods

We used multivariable logistic regression to examine the association of risk factors with severity of COVID-19. We began with a base model that included SNP score, age group and gender. We then included all the candidate variables and used backwards step-wise selection to remove those with $P > 0.05$. We then refined the final model by considering the addition of the removed candidate variables one at a time. Model selection was informed by examination of the Akaike information criterion and the Bayesian information criterion, with a decrease of $> 2$ indicating a statistically significant improvement.

Model calibration was assessed using the Pearson–Windmeijer goodness-of-fit test and model discrimination was measured using the area under the receiver operating characteristic curve (AUC). To compare the effect sizes of the variables in the final model, we used the odds per adjusted standard deviation [20] using dummy variables for age group and ABO blood type. Sensitivity analyses were undertaken by including participants with no hospital records.

We then used the intercept and beta coefficients from the final model to calculate the COVID-19 risk score for all UK Biobank participants.

We used Stata (version 16.1) [21] for analyses; all statistical tests were two-sided and $P < 0.05$ was considered nominally statistically significant.

## Results and discussion

Of the 1,582 UK Biobank participants with a positive SARS-CoV-2 test result and hospital and SNP data available, 564 (35.7%) were from an outpatient setting and considered not to have severe disease (controls), while 1,018 (64.3%) were from an inpatient setting and considered to have severe disease (cases). Cases ranged in age from 51 to 82 years with a mean of 69.1 (standard deviation [SD] = 8.8) years. Controls ranged in age from 50 to 82 years with a mean of 65.0 (SD = 9.0) years. Mean body mass index was 29.0 kg/m$^2$ (SD = 5.4) for cases and 28.5 (SD = 5.4) for controls. Body mass index was transformed to the inverse multiplied by 10 for all analyses and ranged from 0.2 to 0.6 for both cases and controls. The percentage of risk alleles in the SNP score ranged from 47.6 to 73.8 for cases and from 43.7 to 72.5 for controls. The distributions of the variables of interest for cases and controls and the unadjusted odd ratios and 95% confidence intervals (CI) are shown in Table 1.

The adjusted odds ratios for the variables included in the final model are shown in Table 2. This model included SNP score, age group, gender, ethnicity, ABO blood type, and a history of

**Table 1. Characteristics of cases and controls and unadjusted odds ratios for risk of severe COVID-19.**

| Variable | | Cases N = 1018 | Controls N = 564 | Unadjusted odds ratio | 95% confidence interval | P value |
|---|---|---|---|---|---|---|
| **Continuous variables** | | Mean (SD) | Mean (SD) | | | |
| SNP score | % risk alleles | 62.1 (4.1) | 59.3 (4.7) | 1.16 | 1.13 to 1.19 | <0.001 |
| Inverse of body mass index (kg/m$^2$) | 10/BMI | 0.36 (0.06) | 0.36 (0.06) | 0.15 | 0.03 to 0.79 | 0.03 |
| **Categorical variables** | | N (%) | N (%) | | | |
| Age group (years) | 50–59 | 218 (21.4) | 210 (37.2) | – | | |
| | 60–69 | 210 (20.6) | 157 (27.8) | 1.29 | 0.97 to 1.71 | 0.08 |
| | 70+ | 590 (58.0) | 197 (34.9) | 2.89 | 2.25 to 3.70 | <0.001 |
| Gender | Female | 443 (43.5) | 298 (52.8) | – | | |
| | Male | 575 (56.5) | 266 (47.2) | 1.45 | 1.18 to 1.79 | <0.001 |
| Ethnicity | White | 888 (87.2) | 489 (86.7) | – | | |
| | Other | 123 (12.1) | 73 (12.9) | 0.93 | 0.68 to 1.26 | 0.64 |
| | Missing | 7 (0.7) | 2 (0.4) | | | |
| Quintile of Townsend deprivation index at baseline | 1 | 134 (13.2) | 84 (14.9) | – | | |
| | 2 | 165 (16.2) | 95 (16.8) | 1.09 | 0.75 to 1.58 | 0.65 |
| | 3 | 179 (17.6) | 98 (17.4) | 1.14 | 0.79 to 1.65 | 0.47 |
| | 4 | 215 (21.1) | 124 (22.0) | 1.09 | 0.77 to 1.54 | 0.64 |
| | 5 | 325 (31.9) | 162 (28.7) | 1.26 | 0.90 to 1.75 | 0.18 |
| | Missing | 0 (0.0) | 1 (0.2) | | | |
| ABO blood type | O | 425 (41.8) | 235 (41.7) | – | | |
| | A | 450 (44.2) | 249 (44.2) | 1.00 | 0.80 to 1.25 | 1.00 |
| | B | 113 (11.1) | 55 (9.8) | 1.14 | 0.79 to 1.63 | 0.49 |
| | AB | 30 (3.0) | 25 (4.4) | 0.66 | 0.38 to 1.15 | 0.15 |
| Smoking status at baseline | Never/ previous | 882 (86.6) | 499 (88.5) | – | | |
| | Current | 124 (12.2) | 60 (10.6) | 1.17 | 0.84 to 1.62 | 0.35 |
| | Missing | 12 (1.2) | 5 (0.9) | | | |
| Asthma | No | 852 (83.7) | 487 (86.4) | – | | |
| | Yes | 166 (16.3) | 77 (13.7) | 1.23 | 0.92 to 1.65 | 0.16 |
| Autoimmune (rheumatoid arthritis/lupus/ psoriasis) | No | 947 (93.0) | 547 (97.0) | – | | |
| | Yes | 71 (7.0) | 17 (3.0) | 2.41 | 1.41 to 4.14 | 0.001 |
| Cancer–haematological | No | 972 (95.5) | 558 (98.9) | – | | |
| | Yes | 46 (4.5) | 6 (1.1) | 4.40 | 1.87 to 10.37 | 0.001 |
| Cancer–non-haematological | No | 799 (78.5) | 486 (86.2) | – | | |
| | Yes | 219 (21.5) | 78 (13.8) | 1.71 | 1.29 to 2.26 | <0.001 |
| Cerebrovascular disease | No | 847 (83.2) | 503 (89.2) | – | | |
| | Yes | 171 (16.8) | 61 (10.8) | 1.66 | 1.22 to 2.28 | 0.001 |
| Diabetes | No | 765 (75.2) | 493 (87.4) | – | | |
| | Yes | 253 (24.9) | 71 (12.6) | 2.30 | 1.72 to 3.06 | <0.001 |
| Heart disease | No | 633 (62.2) | 437 (77.5) | – | | |
| | Yes | 385 (37.8) | 127 (22.5) | 2.09 | 1.66 to 2.65 | <0.001 |
| Hypertension | No | 419 (41.2) | 354 (62.8) | – | | |
| | Yes | 599 (58.8) | 210 (37.2) | 2.41 | 1.95 to 2.98 | <0.001 |
| Immunocompromised | No | 1,001 (98.3) | 560 (99.3) | – | | |
| | Yes | 17 (1.7) | 4 (0.7) | 2.38 | 0.80 to 7.10 | 0.12 |
| Kidney disease | No | 859 (84.4) | 521 (92.4) | – | | |
| | Yes | 159 (15.6) | 43 (7.6) | 2.24 | 1.57 to 3.20 | <0.001 |

(*Continued*)

**Table 1.** (Continued)

| Variable | | Cases N = 1018 | Controls N = 564 | Unadjusted odds ratio | 95% confidence interval | *P* value |
|---|---|---|---|---|---|---|
| Liver disease | No | 937 (92.0) | 541 (95.9) | – | | |
| | Yes | 81 (8.0) | 23 (4.1) | 2.03 | 1.26 to 3.27 | 0.003 |
| Respiratory disease (excluding asthma) | No | 571 (56.1) | 486 (86.2) | – | | |
| | Yes | 447 (43.9) | 78 (13.8) | 4.88 | 3.73 to 6.38 | <0.001 |

autoimmune disease (rheumatoid arthritis, lupus or psoriasis), haematological cancer, non-haematological cancer, diabetes, hypertension or respiratory disease (excluding asthma) and was a good fit to the data (Windmeijer's H = 0.02, *P* = 0.88). The SNP score was strongly associated with severity of disease, increasing risk by 19% per percentage increase in risk alleles. The effect of age was only evident in the group aged 70 years and over, and while gender was not statistically significant (*P* = 0.26), it was retained because it was one of the three variables considered the base model to which other variables were added. Ethnicity showed a 43% increase in risk for non-whites but was only marginally statistically significant (*P* = 0.06). The AB blood type was protective (*P* = 0.007), but the protective effect of blood type A and the increased risk for blood type B were not statistically significant (*P* = 0.10 and *P* = 0.41,

**Table 2. Final model for risk of severe COVID-19.**

| Variable | | Adjusted odds ratio | 95% confidence interval | *P* value | Odds per adjusted standard deviation | 95% confidence interval |
|---|---|---|---|---|---|---|
| SNP score | % risk alleles | 1.19 | 1.15 to 1.22 | <0.001 | 2.18 | 1.91 to 2.48 |
| Age group (years) | 50–59+ | – | | | | |
| | 60–69 | 0.94 | 0.68 to 1.30 | 0.72 | 0.97 | 0.84 to 1.12 |
| | 70+ | 1.70 | 1.25 to 2.33 | 0.001 | 1.25 | 1.10 to 1.43 |
| Gender | Female | – | | | | |
| | Male | 1.15 | 0.90 to 1.46 | 0.26 | 1.07 | 0.95 to 1.20 |
| Ethnicity | White | – | | | | |
| | Other/ missing | 1.43 | 0.99 to 2.05 | 0.06 | 1.12 | 1.00 to 1.26 |
| ABO blood type | O | – | | | | |
| | A | 0.81 | 0.62 to 1.04 | 0.10 | 0.90 | 0.80 to 1.02 |
| | B | 1.19 | 0.79 to 1.78 | 0.41 | 1.05 | 0.93 to 1.18 |
| | AB | 0.42 | 0.22 to 0.79 | 0.007 | 0.84 | 0.74 to 0.95 |
| Autoimmune disease (rheumatoid arthritis/ lupus/psoriasis) | No | – | | | | |
| | Yes | 2.20 | 1.20 to 4.02 | 0.01 | 1.14 | 1.03 to 1.26 |
| Cancer–haematological | No | – | | | | |
| | Yes | 2.82 | 1.10 to 7.21 | 0.03 | 1.11 | 1.01 to 1.22 |
| Cancer–non-haematological | No | – | | | | |
| | Yes | 1.44 | 1.04 to 2.00 | 0.03 | 1.13 | 1.01 to 1.26 |
| Diabetes | No | – | | | | |
| | Yes | 1.63 | 1.16 to 2.30 | 0.005 | 1.16 | 1.04 to 1.28 |
| Hypertension | No | – | | | | |
| | Yes | 1.35 | 1.03 to 1.78 | 0.03 | 1.13 | 1.01 to 1.26 |
| Respiratory disease (excluding asthma) | No | – | | | | |
| | Yes | 3.43 | 2.54 to 4.64 | <0.001 | 1.48 | 1.35 to 1.63 |

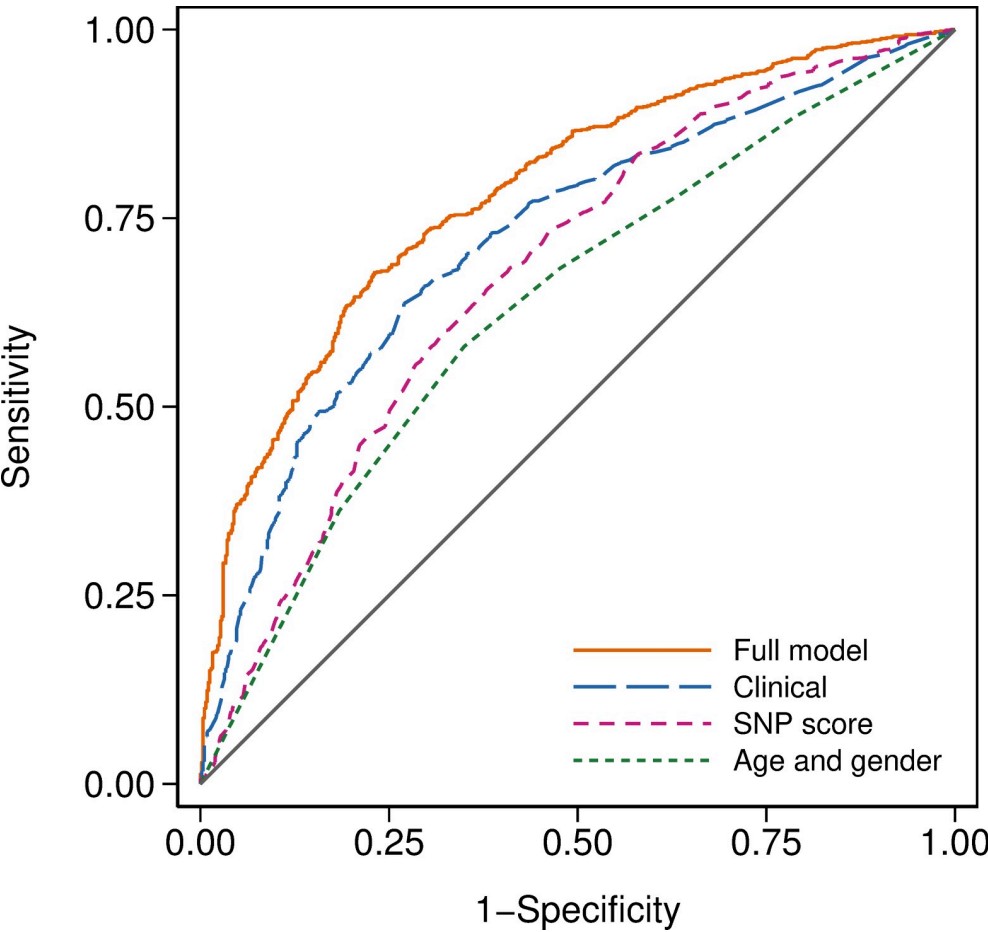

**Fig 1. Receiver operating characteristic curves for models with different amounts of information.** The area under the receiver operating characteristic curve was 0.786 for the full model, 0.723 for the clinical model, 0.680 for the SNP score, and 0.635 for the age and gender model.

respectively). Table 2 also shows the odds per adjusted standard deviation for the final model. This allows direct comparisons of the strength of the associations for each variable, regardless of the scales on which they were measured. The SNP score was, by far, the strongest predictor followed by respiratory disease and age 70 years or older. Sensitivity analyses including those with no linked hospital records did not change the conclusions presented in S3 Table.

The receiver operating characteristic curves for the final model and for alternative models with clinical factors only (S4 Table); SNP score only (Table 1); and age and gender (S5 Table) are shown in Fig 1. The SNP score alone had an AUC of 0.680 (95% CI, 0.652 to 0.708). The model with age and gender had an AUC of 0.635 (95% CI, 0.607 to 0.662), while the model with clinical factors only had an AUC of 0.723 (95% CI, 0.698 to 0.749). Given that the minimum possible value for an AUC is 0.5, the model with clinical factors only was a 65% improvement over the model with age and gender ($\chi^2$ = 57.97, df = 1, $P$<0.001). The full model had an AUC of 0.786 (95% CI, 0.763 to 0.808) and was an 28% improvement over the model with clinical factors only ($\chi^2$ = 39.54, df = 1, $P$<0.001), a 59% improvement over the SNP score ($\chi^2$ = 71.94, df = 1, $P$<0.001), and a 111% improvement over the model with age and gender ($\chi^2$ = 113.67, df = 1, $P$<0.001).

**A** Cases

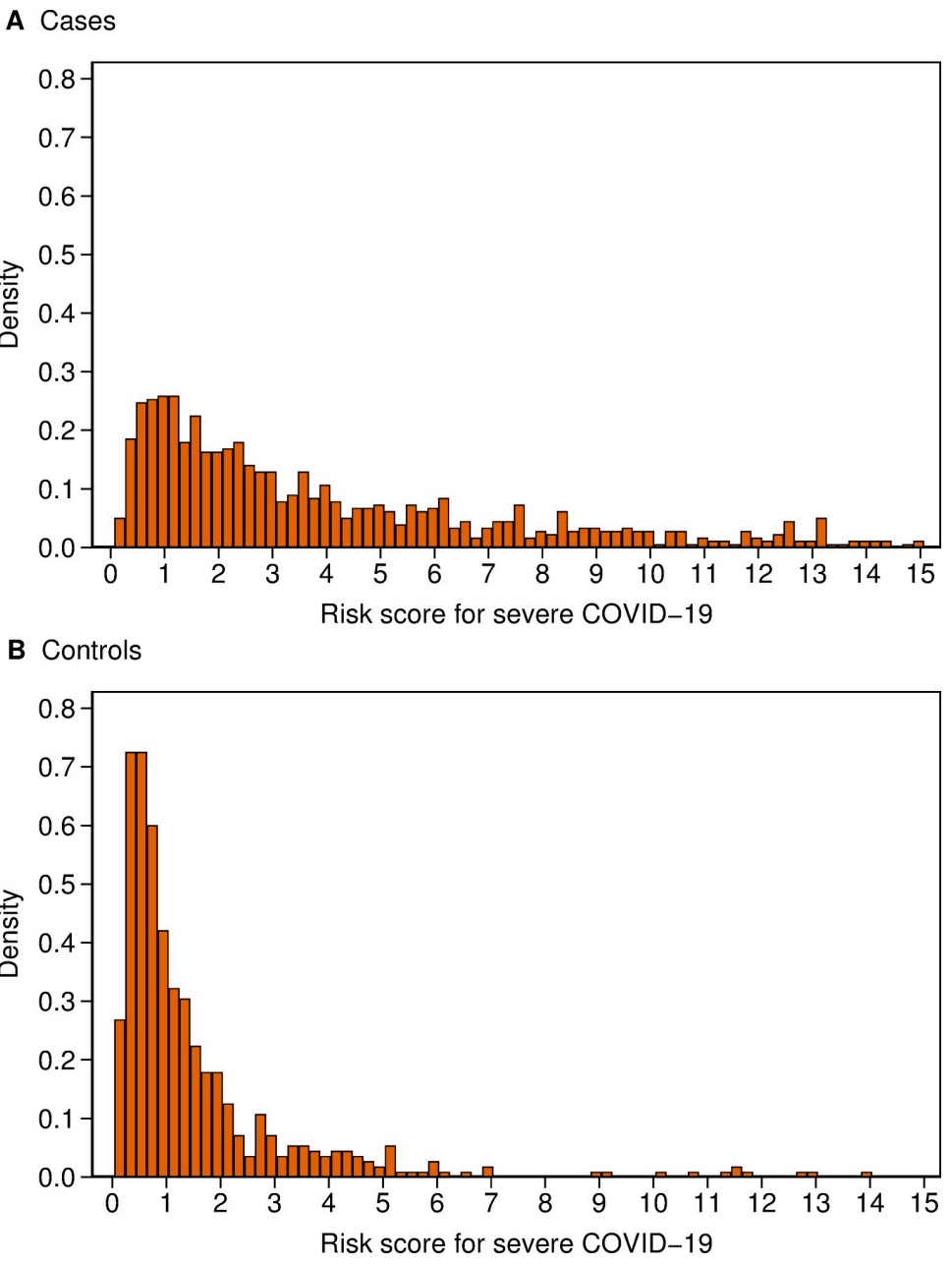

**B** Controls

**Fig 2.** Distribution of risk score for severe COVID-19 risk score for (A) cases and (B) controls. Note that 130 (13%) cases and 6 (1%) controls with scores of 15 or over have been omitted to facilitate the display of the distribution.

Fig 2 illustrates the difference in the distributions of the COVID-19 risk scores in cases and controls. The median score was 3.35 for cases and 0.90 for controls, with inter-quartile ranges of 6.70 and 1.34, respectively. Sixteen per cent of cases and 53% of controls had COVID-19 risk scores of less than 1, and 18% of cases and 25% of controls had scores $\geq$1 and $<$2. COVID-19 risk scores $\geq$2 were more common in cases than in controls, with 13% of cases and 9% of controls having scores $\geq$2 and $<$3, 8% of cases and 4% of controls having scores $\geq$3 and $<$4, and 45% of cases and 9% of controls having scores $\geq$4.

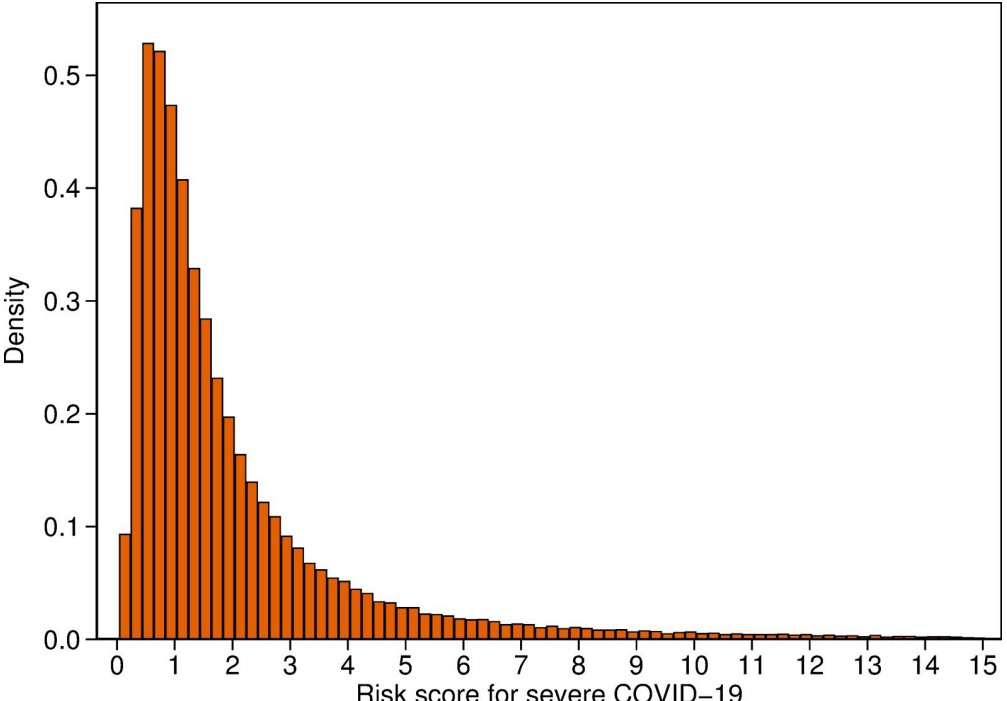

**Fig 3. Distribution of risk score for severe COVID-19 in the 487,311 UK Biobank participants with SNP data available.** Note that 7,769 (1.8%) scores of 15 or over have been omitted to facilitate the display of the distribution.

Fig 3 shows that the distribution of the COVID-19 risk score in the whole UK Biobank is similar to that for the controls in Fig 2B. The median COVID-19 risk score in the whole UK Biobank was 1.32 and the inter-quartile range was 1.80. Thirty-eight per cent of the UK Biobank have COVID-19 risk scores of less than 1, while 29% have scores $\geq$1 and <2, 13% have scores $\geq$2 and <3, 6% have scores $\geq$3 and <4, and 14% have scores of $\geq$4.

One of the main issues of the COVID-19 pandemic is that of susceptibility to severe disease. We have shown that a comprehensive risk prediction test that quantifies the varying effects of clinical risk factors and a SNP risk score has an AUC of 0.786 and improves risk discrimination of severe COVID-19 by 111% compared with a model using age and gender ($P$<0.001). Examination of the odds per adjusted standard deviation (Table 2) shows that the SNP score is the strongest risk factor for severe COVID-19. While the SNP score explains more variance in disease severity than all of the other risk factors in the model combined, the full model discriminates better than the clinical factors alone or the SNP score alone (both $P$<0.001).

The strong associations observed in the model consisting of just age and gender (S4 Table) are attenuated by the inclusion of other risk factors. This is due to the comorbidities in the full model being more prevalent in older people and in men, and it is the comorbidities–not age and gender–that are associated with severe disease. Relying on age and gender alone to determine risk of severe COVID-19 will unnecessarily classify healthy older people as being at high risk and will fail to accurately quantify the increased risk for younger people with comorbidities.

Our study does have some limitations. We used source of test result as a proxy for severity of disease. Therefore, there may have been some misclassification of disease severity, but this would be likely to attenuate the magnitude of the associations. Townsend deprivation score, BMI and current smoking status were taken from the baseline assessment data and may not

represent the participants' current status. This may have contributed to these variables not being statistically significant. Until mid-May, testing for COVID-19 in the UK was limited to those who had recognisable symptoms and were essential workers, contacts of known cases, hospitalised or had returned from overseas [22]. Therefore, many asymptomatic or very mild cases from the first wave of the pandemic will not have been identified in this dataset. Nevertheless, our results remain applicable to those who develop symptoms that warrant medical attention.

## Conclusions

While the vast majority of the 487,311 UK Biobank participants with SNP data available are at low or only slightly elevated risk of severe COVID-19 (Fig 3), we can identify those who are likely to be at substantially increased risk. Our risk prediction test for severe COVID-19 in people aged 50 years or older has great potential for wide-reaching benefits in managing the risk for essential workers, in healthcare settings and in workplaces that seek to operate safely. The test will also enable individuals to make informed choices based on their personal risk. However, key to understanding the performance of our risk prediction test will be validation in independent data sets, work that we are planning to undertake in the near future.

## Supporting information

**S1 Table. Single-nucleotide polymorphisms.**
(PDF)

**S2 Table. Disease definitions.**
(PDF)

**S3 Table. Sensitivity analysis.**
(PDF)

**S4 Table. Model with age group and gender.**
(PDF)

**S5 Table. Model with clinical risk factors.**
(PDF)

## Acknowledgments

We wish to thank Mr Lawrence Whiting for his invaluable expertise in the management of large data files from the UK Biobank.

## Author Contributions

**Conceptualization:** Gillian S. Dite, Nicholas M. Murphy, Richard Allman.

**Data curation:** Gillian S. Dite.

**Formal analysis:** Gillian S. Dite.

**Investigation:** Gillian S. Dite, Nicholas M. Murphy, Richard Allman.

**Methodology:** Gillian S. Dite, Nicholas M. Murphy, Richard Allman.

**Project administration:** Nicholas M. Murphy, Richard Allman.

**Software:** Gillian S. Dite.

**Writing – original draft:** Gillian S. Dite, Nicholas M. Murphy, Richard Allman.

**Writing – review & editing:** Gillian S. Dite, Nicholas M. Murphy, Richard Allman.

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
