## [Decision Letter · Decision Letter 0]

30 Dec 2020

PONE-D-20-34815

An integrated clinical and genetic model for predicting risk of severe COVID-19: A population-based case–control study

PLOS ONE

Dear Dr. Dite,

Thank you for submitting your manuscript to PLOS ONE. After careful consideration, we feel that it has merit but does not fully meet PLOS ONE’s publication criteria as it currently stands. Therefore, we invite you to submit a revised version of the manuscript that addresses the points raised during the review process.

We look forward to receiving your revised manuscript.

Kind regards,

Giuseppe Novelli

Academic Editor

PLOS ONE

Journal Requirements:

"I have read the journal's policy and the authors of this manuscript have the following competing interests: All authors are employed by Genetic Technologies Limited and have a patent pending for the work in this manuscript."

We note that one or more of the authors are employed by a commercial company: Genetic Technologies Limited .

2.1. Please provide an amended Funding Statement declaring this commercial affiliation, as well as a statement regarding the Role of Funders in your study. If the funding organization did not play a role in the study design, data collection and analysis, decision to publish, or preparation of the manuscript and only provided financial support in the form of authors' salaries and/or research materials, please review your statements relating to the author contributions, and ensure you have specifically and accurately indicated the role(s) that these authors had in your study. You can update author roles in the Author Contributions section of the online submission form.

2.2. Please also provide an updated Competing Interests Statement declaring this commercial affiliation along with any other relevant declarations relating to employment, consultancy, patents, products in development, or marketed products, etc.  

2.3. We note that you have a patent relating to material pertinent to this article. Please provide an amended statement of Competing Interests to declare this patent (with details including name and number), along with any other relevant declarations relating to employment, consultancy, patents, products in development or modified products etc. Please confirm that this does not alter your adherence to all PLOS ONE policies on sharing data and materials, as detailed online in our guide for authors http://journals.plos.org/plosone/s/competing-interests by including the following statement: "This does not alter our adherence to  PLOS ONE policies on sharing data and materials.” If there are restrictions on sharing of data and/or materials, please state these. Please note that we cannot proceed with consideration of your article until this information has been declared.

2.4. Please know it is PLOS ONE policy for corresponding authors to declare, on behalf of all authors, all potential competing interests for the purposes of transparency. PLOS defines a competing interest as anything that interferes with, or could reasonably be perceived as interfering with, the full and objective presentation, peer review, editorial decision-making, or publication of research or non-research articles submitted to one of the journals. Competing interests can be financial or non-financial, professional, or personal. Competing interests can arise in relationship to an organization or another person. Please follow this link to our website for more details on competing interests: http://journals.plos.org/plosone/s/competing-interests

Reviewers' comments:

Reviewer's Responses to Questions

**Comments to the Author**

1. Is the manuscript technically sound, and do the data support the conclusions?

Reviewer #1: Partly

Reviewer #2: Yes

2. Has the statistical analysis been performed appropriately and rigorously? 

Reviewer #1: Yes

Reviewer #2: Yes

3. Have the authors made all data underlying the findings in their manuscript fully available?

Reviewer #1: Yes

Reviewer #2: No

4. Is the manuscript presented in an intelligible fashion and written in standard English?

Reviewer #1: Yes

Reviewer #2: Yes

5. Review Comments to the Author

Reviewer #1: I was pleased to see this paper. Very crucial topic.

My only concerns are about the selection of the SNPs for the score, quality controls procedures and imputation of the data. Could the authors give more details about these points?

If I understand well from the paragraph "We used the

126 percentage rather than a count because some of the eligible participants had missing data for

127 some SNPs (9% had all SNPs genotyped, 82% were missing 1–5 SNPs and 9% were missing

128 6–15 SNPs)" the number of particpants with all the SNPs genotyped is quite low.

How did the authors manage this point? Did the authors use an imputation method?

Reviewer #2: The principal aim of this case-control study is to prospect a new and more accurate prediction model of severe COVID-19, based on age, gender, clinical conditions and a SNP score built on a panel of 64 SNPs identified from published data.

The authors used the clinical baseline data collected at the UK Biobank for 1,582 participants aged >50 years, tested positive for COVID-19. These individuals were divided into two categories: severe disease (n=1,018) and without severe disease (n=564). They identified 62 SNPs from a meta-analysis conducted by the COVID-19 host Genetic Initiative consortium and two SNPs from the loci found by Ellinghaus et al., for a total of 64 SNPs. Clinical risk factors were identified on large epidemiological studies of electronic health records (rare monogenic diseases were not included). Multivariable logistic regression was used to examine the association of risk factors and severe COVID-19 with adjusted odds ratio for each variable.

The results showed a strong association between the SNP score and the severity of the disease, while gender was not statistically significant (p=0.26) and the effect of age was relevant only for >70 years.

Surely the COVID-19 pandemic has set the need for a more accurate and complete risk prediction model than the one based just on age, gender and comorbidities. The addition of genetic risk factors could provide useful information to the community, especially in terms of prevention (more than a triage/hospital setting; lines 58-59). However, we should bear in mind that an accurate prediction model in order to be useful should also be easily applicable to the community, in terms of availability and cost-effectiveness (therefore this kind of data should be available for every individual, including SNPs genotype). Even though I share the authors approach to go beyond age and gender to assess individual’s risk for COVID-19, analysing this study I personally found some weaknesses, for example:

As they point out themselves (line 257), the main limitation of this study relies in the given meaning to the source of test results. We can assume that outpatients did not present a severe form of disease at time of testing, but what about the prognosis? The same consideration applies to inpatients: surely not everyone of them developed a severe form of COVID-19 and some of them could have been hospitalized for quite something else. Consequentially, can we really talk about risk prediction of “severe” COVID-19? Moreover, as stated in lines 274-275, the prediction model should be tested in independent data sets to confirm its reliability.

As we are considering the percentage of risk alleles, it should be taken into account that the 64 SNPs are not known for every patient: only 9% has all SNPs genotyped, 9% of them has between 6 and 15 SNPs missing and the rest has 1-5 SNPs missing.

Concerning primary care data (Lines 90-91): maybe the authors could specify which kind of clinical information was not taken into account.

Tables ad images are well done and easy to read, even though I would have specified for figure 3, which should refer to the “vast majority of UK Biobank participants” (line 268), how many participants without SARS-CoV-2 test results were actually taken into consideration (since we know that among them only 18,221 had SARS-CoV-2 test results; line 95).

6. PLOS authors have the option to publish the peer review history of their article (what does this mean?). If published, this will include your full peer review and any attached files.

Reviewer #1: No

Reviewer #2: No

---

## [Author Response · Author response to Decision Letter 0]

7 Jan 2021

The information below has also been provided in the Response to Reviewers document.

Reviewer #1: 

Question: 

I was pleased to see this paper. Very crucial topic.

My only concerns are about the selection of the SNPs for the score, quality controls procedures and imputation of the data. Could the authors give more details about these points?

Response:

The GWAS quality control, imputation and meta-analysis, were performed by the COVID-19 Host Genetics Initiative consortium, details of which are available from the references provided.

We have added additional details of our SNP selection on lines 105–109, which now reads:

We identified 62 SNPs from the results of the ANA2 meta-analysis (release 2) of SARS-CoV2 positive non-hospitalised versus hospitalised cases of COVID-19 conducted by the COVID-19 Host Genetics Initiative consortium [8, 9]. Because of the limited amount of data available at the time of release, we used P<0.0001 as the threshold for loci selection. We then removed and variants that were associated with hospitalisation in only one of the five studies in the meta-analysis.

We have also added the following sentence on line 114–115:

All of the variants had a Cochran’s Q heterogeneity test P>0.001.

Question: 

If I understand well from the paragraph "We used the percentage rather than a count because some of the eligible participants had missing data for some SNPs (9% had all SNPs genotyped, 82% were missing 1–5 SNPs and 9% were missing 6–15 SNPs)" the number of particpants with all the SNPs genotyped is quite low.

How did the authors manage this point? Did the authors use an imputation method?

Response:

We did not impute the missing SNPs because, while only 9% of participants had all SNPs genotyped, very few participants had more than 5 SNPs missing. 

Using a percentage rather than a count of risk alleles ensures that the SNP score for those with missing data does not underestimate risk. Because the SNPs will be missing completely at random, missing SNP data will not bias the calculation of the percentage.

Nevertheless, we have updated the information in the parentheses (lines 127–129) to be more informative. It now reads:

9% had all SNPs genotyped, 21% were missing 1 SNP, 26% were missing 2 SNPs, 18% were missing 3 SNPs, 10% were missing 4 SNPs, 7% were missing 5 SNPs, 8% were missing 6–10 SNPs and 1% were missing 11–15 SNPs

Reviewer #2: 

Question:

The principal aim of this case-control study is to prospect a new and more accurate prediction model of severe COVID-19, based on age, gender, clinical conditions and a SNP score built on a panel of 64 SNPs identified from published data.

The authors used the clinical baseline data collected at the UK Biobank for 1,582 participants aged >50 years, tested positive for COVID-19. These individuals were divided into two categories: severe disease (n=1,018) and without severe disease (n=564). They identified 62 SNPs from a meta-analysis conducted by the COVID-19 host Genetic Initiative consortium and two SNPs from the loci found by Ellinghaus et al., for a total of 64 SNPs. Clinical risk factors were identified on large epidemiological studies of electronic health records (rare monogenic diseases were not included). Multivariable logistic regression was used to examine the association of risk factors and severe COVID-19 with adjusted odds ratio for each variable.

The results showed a strong association between the SNP score and the severity of the disease, while gender was not statistically significant (p=0.26) and the effect of age was relevant only for >70 years.

Surely the COVID-19 pandemic has set the need for a more accurate and complete risk prediction model than the one based just on age, gender and comorbidities. The addition of genetic risk factors could provide useful information to the community, especially in terms of prevention (more than a triage/hospital setting; lines 58-59). However, we should bear in mind that an accurate prediction model in order to be useful should also be easily applicable to the community, in terms of availability and cost-effectiveness (therefore this kind of data should be available for every individual, including SNPs genotype). Even though I share the authors approach to go beyond age and gender to assess individual’s risk for COVID-19, analysing this study I personally found some weaknesses, for example:

As they point out themselves (line 257), the main limitation of this study relies in the given meaning to the source of test results. We can assume that outpatients did not present a severe form of disease at time of testing, but what about the prognosis? The same consideration applies to inpatients: surely not everyone of them developed a severe form of COVID-19 and some of them could have been hospitalized for quite something else. Consequentially, can we really talk about risk prediction of “severe” COVID-19? Moreover, as stated in lines 274-275, the prediction model should be tested in independent data sets to confirm its reliability.

Response:

The UK Biobank did not have extensive clinical information available to determine severity of disease. We have explained that we considered our outcome measure to be a proxy for severity of disease. If there were to be some misclassification of severity of disease, this would bias the results towards the null.

The data file provided by the UK Biobank had, in many cases, multiple test results for participants. We aggregated the data so that participants who tested positive both in an outpatient setting and in hospital were considered to have severe disease (see lines 97–100). This addresses (to some extent) the problem of prognosis of disease raised by the reviewer. We acknowledge that we cannot identify participants who were hospitalised for reasons other than COVID-19.

In an emerging health crisis such as COVID-19, the perfect dataset does not exist and we must use data that is readily available. Our concerns regarding the classification of disease severity and the limited testing available early in the pandemic (the period from which this data was derived) were behind our decision not to divide the data into training and testing datasets. We believed that any inherent biases in the data would result in false validation of the risk prediction model. This is why we emphasised the need for validation in independent datasets.

Question: 

As we are considering the percentage of risk alleles, it should be taken into account that the 64 SNPs are not known for every patient: only 9% has all SNPs genotyped, 9% of them has between 6 and 15 SNPs missing and the rest has 1-5 SNPs missing.

Please see the response above to the question from Reviewer 1.

Question: 

Concerning primary care data (Lines 90-91): maybe the authors could specify which kind of clinical information was not taken into account.

Response:

We have updated the text on lines 85–87 as follows:

At the time of data extraction, primary care administrative data (general practitioner records relating to diagnoses, symptoms, referrals, laboratory test results and prescriptions for medication) was only available for just over half of the identified participants and was therefore not used in these analyses.

Question: 

Tables ad images are well done and easy to read, even though I would have specified for figure 3, which should refer to the “vast majority of UK Biobank participants” (line 268), how many participants without SARS-CoV-2 test results were actually taken into consideration (since we know that among them only 18,221 had SARS-CoV-2 test results; line 95).

Response:

On line 272, “the vast majority of UK Biobank participants” refers to those with low or only slightly increased risk among the whole UK Biobank, not the subset with COVID-19 test results in the main analyses.

We have updated the text on lines 272–273 as follows:

While the vast majority of the 487,311 UK Biobank participants with SNP data available are at low or only slightly elevated risk of severe COVID-19 (Fig 3)…

We have also updated the caption to Fig 3 (lines 240–241) as follows:

Fig 3. Distribution of risk score for severe COVID-19 in the 487,311 UK Biobank participants with SNP data available.

---

## [Editor Report · Decision Letter 1]

3 Feb 2021

An integrated clinical and genetic model for predicting risk of severe COVID-19: A population-based case–control study

PONE-D-20-34815R1

Dear Dr. Dite,

We’re pleased to inform you that your manuscript has been judged scientifically suitable for publication and will be formally accepted for publication once it meets all outstanding technical requirements.

Kind regards,

Giuseppe Novelli

Academic Editor

PLOS ONE
---

## [Editor Report · Acceptance letter]

5 Feb 2021

PONE-D-20-34815R1 

An integrated clinical and genetic model for predicting risk of severe COVID-19: A population-based case–control study 

Dear Dr. Dite:

I'm pleased to inform you that your manuscript has been deemed suitable for publication in PLOS ONE. Congratulations! Your manuscript is now with our production department. 

Kind regards, 

on behalf of

Prof. Giuseppe Novelli 

Academic Editor

PLOS ONE